# Pain Relieving Effect of-NSAIDs-CAIs Hybrid Molecules: Systemic and Intra-Articular Treatments against Rheumatoid Arthritis

**DOI:** 10.3390/ijms20081923

**Published:** 2019-04-18

**Authors:** Laura Micheli, Murat Bozdag, Ozlem Akgul, Fabrizio Carta, Clizia Guccione, Maria Camilla Bergonzi, Anna Rita Bilia, Lorenzo Cinci, Elena Lucarini, Carmen Parisio, Claudiu T. Supuran, Carla Ghelardini, Lorenzo Di Cesare Mannelli

**Affiliations:** 1Department of Neuroscience, Psychology, Drug Research and Child Health-NEUROFARBA-Pharmacology and Toxicology Section, University of Florence, Viale Gaetano Pieraccini 6, 50139 Florence, Italy; laura.micheli@unifi.it (L.M.); lorenzo.cinci@unifi.it (L.C.); elena.lucarini@unifi.it (E.L.); carmen.parisio@unifi.it (C.P.); carla.ghelardini@unifi.it (C.G.); 2Department of Neuroscience, Psychology, Drug Research and Child’s Health (NEUROFARBA), Section of Pharmaceutical and Nutraceutical Sciences, University of Florence, Via Ugo Schiff 6, Sesto Fiorentino, 50019 Florence, Italy; bozdag.murat@unifi.it (M.B.); fabrizio.carta@unifi.it (F.C.); claudiu.supuran@unifi.it (C.T.S.); 3Faculty of Pharmacy, Department of Pharmaceutical Chemistry, Ege University Bornova, 35100 Izmir, Turkey; ozlem.akgul@ege.edu.tr; 4Department of Chemistry, University of Florence, Via U. Schiff 6, Sesto Fiorentino, 50019 Florence, Italy; clizia.guccione@unifi.it (C.G.); mariacamilla.bergonzi@unifi.it (M.C.B.); annarita.bilia@unifi.it (A.R.B.)

**Keywords:** carbonic anhydrase inhibitors (CAIs), dual hybrids, inflammation, hypersensitivity, rheumatoid arthritis, CFA

## Abstract

To study new target-oriented molecules that are active against rheumatoid arthritis-dependent pain, new dual inhibitors incorporating both a carbonic anhydrase (CA)-binding moiety and a cyclooxygenase inhibitor (NSAID) were tested in a rat model of rheumatoid arthritis induced by CFA intra-articular (i.a.) injection. A comparison between a repeated *per os* treatment and a single i.a. injection was performed. CFA (50 µL) was injected in the tibiotarsal joint, and the effect of *per os* repeated treatment (1 mg kg^−1^) or single i.a injection (1 mg mL^−1^, 50 µL) with NSAIDs-CAIs hybrid molecules, named **4** and **5**, was evaluated. The molecules **4** and **5**, which were administered daily for 14 days, significantly prevented CFA-induced hypersensitivity to mechanical noxious (Paw pressure test) and non-noxious stimuli (von Frey test), the postural unbalance related to spontaneous pain (Incapacitance test) and motor alterations (Beam balance test). Moreover, to study a possible localized activity, **4** and **5** were formulated in liposomes (lipo **4** and lipo **5**, both 1 mg mL^−1^) and directly administered by a single i.a. injection seven days after CFA injection. Lipo **5** decreased the mechanical hypersensitivity to noxious and non-noxious stimuli and improved motor coordination. Oral and i.a. treatments did not rescue the joint, as shown by the histological analysis. This new class of potent molecules, which is able to inhibit at the same time CA and cyclooxygenase, shows high activity in a preclinical condition of rheumatoid arthritis, strongly suggesting a novel attractive pharmacodynamic profile.

## 1. Introduction

Rheumatoid arthritis (RA) is a chronic inflammatory joint disease that is characterized by synovial proliferation, joint destruction, and systemic inflammation [1,2], and it affects about 1–2% of the global adult population [3]. Although the autoimmune reaction remains clinically silent, inflammation has the potential to cause severe pain and swelling.

Despite the wide range of available treatments, rheumatoid arthritis (RA) continues to pose a substantial burden across the globe, with unmet needs in key domains, such as pain, physical and mental functions, and fatigue. The conventional strategy of medical therapy for RA involves the nonsteroidal anti-inflammatory drugs (NSAIDs) [4], but the side effect and resistance of these pharmaceuticals usually cause chronic pain and poor life quality and they require a larger amount of resources for further management [5]. Other classes of drugs include the disease-modifying antirheumatic drugs (DMARDs), biologics, JAK-inhibitors, and corticosteroids, or they can be used a combination of these drug classes along with the recommendation to conduct proper physical activities, with the intent to tackle the symptoms as well as the progression of the pathology [6,7].

Recent publications demonstrated that several isoforms of carbonic anhydrase (CAs, EC 4.2.1.1), metalloenzymes, which reversibly catalyse the conversion of CO_2_ to the bicarbonate ions and protons, are involved in articular diseases, such as CA I, which is over-expressed in the synovium of the patients with ankylosing spondylitis [8]. Moreover, transgenic mice over-expressing CA I showed aggravated joint inflammation and destruction [9], and antibodies to CA III and IV have been identified in rheumatoid arthritis [10]. Recently, Margheri and colleagues [11] highlighted the overexpression of the transmembrane carbonic anhydrase isoform IX and XII in the synovium of patients that are affected by juvenile idiopathic arthritis, the most common form of chronic rheumatic disease, which affects children worldwide and shares several features with adult RA [11] These evidences led us to consider CA as a promising new target for the treatment of articular pain. Indeed, we recently reported two studies demonstrating that a single administration of hybrid molecules composed by the nonsteroidal-anti-inflammatory drugs and carbonic-anhydrase inhibitors (NSAIDs-CAIs) was more effective in comparison to the NSAIDs alone in a rat model of rheumatoid arthritis that is induced by CFA [12,13]. 

In the present study, we deeply investigated the anti-hyperalgesic effect of two hybrid NSAIDs-CAIs molecules in a rat model of CFA-induced rheumatoid arthritis. In particular, we compared the efficacy of a repeated *per os* treatment with respect to a single intra-articular injection. To increase the solubility, the hybrids were reformulated in liposomes for the i.a. administration. The liposomes are composed of bilayer of phospholipids with an aqueous reservoir and they allow the encapsulation of a wide variety of hydrophilic and hydrophobic agents. They are selected as the drug delivery system for NSAIDs-CAIs molecules, because the phospholipid bilayer structure similar to physiological membranes makes them highly biocompatible, biodegradable, and non-toxic. Moreover, the histological analysis of joints was conducted after both treatments to highlight a possible protective mechanism.

## 2. Results

In a previous study, we demonstrated the acute pain relieving effects of nonsteroidal anti-inflammatory drugs and carbonic inhibitors (NSAIDs-CAIs) hybrids in a rat model of CFA-induced rheumatoid arthritis [13]. Among the series, Compounds **4** and **5** (CAIs inhibitors linked with sulindac and diclofenac, respectively) highlighted the best anti-hypersensitivity profile [13] (Scheme 1).

Both of the compounds were prepared according to the procedure that was described by [13]. A carboxylic acid NSAID derivative (i.e., sulindac 2 or diclofenac 3) was treated with *N*-hydroxysuccinimide (NHS) and *N*-(3-dimethylaminopropyl)-*N*′-ethylcarbodiimide hydrochloride (EDCl) in dry *N*,*N*-dimethylformamide (DMF) to obtain activated esters. Subsequently, the commercially avaliable 4-(2-amino-ethyl)benzene sulphonamide 1 (CAI) was added to perform in situ the amide coupling reactions in order to obtain the desired CAI-NSAID hybrid molecules 4 and 5, after the appropriated purification method was carried out [13]. Moreover for compounds 4 and 5, the inhibition properties against the relevant hCA isoforms (i.e., I, II, IV, IX, and XII) while using the stopped-flow carbon dioxide hydration assay are reported in Table 1.

As extension of our study, herein we report an in silico evaluation of the logP values of such compounds in order to gather information regarding their lipophilic properties. The predictions were carried out using the Swiss ADME program that is available on-line [14,15], which calculates logPs by means of five different methods and their corresponding values are all merged into the consensus logP (clogP) parameter that was obtained as the mathematical mean of all five predictions. The NSAID-CAI derivatives **4** and **5** showed calculated clogPs of 4.20 and 3.97, respectively, thus within the logP range of 2–5, which is optimal for drugs that are intended for systemic administration. 

Therefore, we decided to deeply investigate their efficacy after a repeated treatment in the same model of articular pain. The compounds were orally administered daily at the dose of 1 mg kg^−1^, starting from the same day of CFA intra-articular injection. The dose was chosen on the basis of previous experiments [13]. Behavioural measurements were performed on days 7 and 14, 30 min and 24 h after the last treatment (Figure 1). 

The CFA + vehicle treated animals tolerated, on the ipsilateral paw, a weight of 43.0 ± 0.5 g and 43.3 ± 1.7 g, in comparison to 63.3 ± 0.8 g and 65.0 ± 1.4 g of the control group on days 7 and 14, respectively (Figure 1). Repeated daily treatment with **4** was able to increase the ipsilateral paw threshold up to 56.7 ± 2.3 g on day 7, 24 h after the last treatment without increasing the analgesic efficacy 30 min after the new administration. On day 14, the anti-hypersensitivity effect of this compound decreased when the measure was performed 24 h after the last treatment (49.2 ± 2.2 g), but it showed an enhancement at 30 min. The compound **5** (daily administered at the dose of 1 mg kg^−1^) showed a lower efficacy with respect to the molecule **4** (Figure 1). It was slightly effective in reducing CFA-induced mechanical hypersensitivity 24 h after the last administration both on days 7 and 14. On the contrary, we highlighted an increase of its efficacy when the measures were performed at 30 min (Figure 1). Values that were measured on the contralateral paw did not show **4** and **5** activities on the normal pain sensitivity. The von Frey test, employing a mechanical stimulus that does not normally provoke pain, evaluated the pain threshold (Figure 2). 

The ipsilateral paw withdrawal threshold of the CFA + vehicle treated rats decreased to 10.3 ± 0.4 g on day 7 and to 12.1 ± 1.0 g on day 14 as compared to the vehicle + vehicle treated group (27.4 ± 0.4 g on day 7 and 26.0 ± 2.2 g on day 14). Four repeated treatments significantly reduced CFA-induced hypersensitivity with non-progressively increasing efficacy during a time when the measurements were performed 24 h after the last treatment (15.6 ± 0.6 g on day 7 and 16.5 ± 0.4 g on day 14). An enhancement of its efficacy was highlighted at 30 min after the daily administration. Compound **5** was effective only at 30 min both on days 7 and 14 (Figure 2). The values measured on the contralateral paw did not show **4** and **5** activity on the normal pain threshold.

Moreover, we evaluated motor coordination—primarily balance and proprioception using the Beam Balance test—providing a score to each animal when it walked on the beam. The ability of the animals to balance and traverse on a beam was significantly worse for CFA + vehicle group at 7 and 14 days after CFA injection with respect to the vehicle + vehicle treated rats (Figure 3). 

Four treatments significantly increased the performance of the animals, improving the ability to traverse the beam, and increasing balance and proprioception on days 7 and 14. The compound **5** was only effective on day 7, 30 min after administration (Figure 3). The effect of **4** and **5** on morphological derangement of the tibiotarsal joint was evaluated in the animals after 14 days of repeated treatments. In CFA + vehicle treated animals, the joint space was completely replaced by fibrous tissue, showing several collagen fibers that were stained by eosin. Moreover, the presence of abundant inflammatory infiltrate characterized fibrosis. Animals that were treated with compounds showed histological features that were comparable to those that are described in CFA treated animals (Panel S1). Additionally, it is worth noting the presence, in CFA treated animals, of few cartilage and bone erosion foci. In fact, areas in which cartilage and bone appeared rarefied were visible along the edges of fibrotic tissue. The treatment with compounds (**4** and **5**) did not exert effects on cartilage and bone erosion. Quantitative results of morphological parameters score were reported in Panel S1.

In parallel to the oral treatment, the intra-articular joint injection is a common therapeutic procedure that aims to reduce pain and inflammation. We then evaluated the pain relief effect of this compounds after intra-articular injection. Compounds **4** and **5** were formulated in liposomes (lipo **4** and lipo **5**, respectively) to prepare a solution for a parenteral administration. All of the liposomal dispersions were analysed in terms of size (nm), PDI and ζ-potentials by photon correlation spectroscopy; Table 1 summarizes the data obtained. After these results, nanocarriers loaded with 1 mg mL^−1^ of molecules were selected to be tested *in vivo*. The mean diameters of the chosen loaded-vesicles were ≤250 nm and resulted in being suitable for systemic administration. PDI is a dimensionless measure of the broadness of size distribution; these measurements were performed three-times for each formulation and they were satisfactorily reproducible. ζ-Potential was around −20 mV for all the prepared formulations. The **4** loading efficiency was about 30%, while the EE% was higher, around 75%, for **5** loaded liposomes, due to the amphiphilic structure of the molecule (Appendix A). 

Chemical and physical stability in storage condition of both loaded-vesicle dispersions was monitored over 15 days. Physical stability was proved monitoring size and polydispersity by Dynamic Light Scattering (DLS) measurements. Size of nanocarriers were maintained during this period (Appendix A) Moreover, the homogeneity of all the liposomal formulations were stable during time (Appendix A). According to these results, no vesicle size alterations happened until the end of the *in vivo* investigations. Additionally, chemical stability was checked by the quantification of drug at 5, 10, and 15 days by HPLC PDA analyses. After 15 days, the initial amount of molecules was reduced by 1.6% for 4 and 1.4% for 5 (Appendix A). At the end of the physical and chemical stability studies, both of the formulations were well acceptable during the first week. However, the solutions of all nanoformulations were freshly prepared for *in vivo* tests.

Fifty µL of lipo **4** and lipo **5** were i.a. injected into the tibio-tarsal joint seven days after CFA injection. Behavioural measurements that were performed on days 14 and 21 showed that only lipo **5** was able to significantly increase the weight that the animals can tolerate on the ipsilateral paws with respect to the CFA + vehicle group (Figure 4). 

Lipo **4** was ineffective. Similar results were obtained measuring the response to a non-noxious mechanical stimulus with the von Frey test (Figure 5).

Additionally, while evaluating the motor skills of the animals, we highlighted an improvement regarding the balance and the capacity to traverse the beam of the animals that were treated with lipo **5** with respect to the CFA group. Lipo **4** was ineffective (Figure 6).

The histological evaluations were performed on day 21 from the beginning of the experiment. Animals treated with lipo **4** and lipo **5** showed histological features that were comparable to those described in CFA treated animals (Panel S2). The treatment with compounds (lipo **4** and lipo **5**) did not exert effects on cartilage and bone erosion (Appendix A). Appendix A reports the quantitative results of morphological parameters score.

## 3. Discussion

This study reports the preclinical efficacy of NSAIDs-CAIs hybrid molecules in relieving pain sensitivity that is induced by CFA i.a. injection in rats. A comparison between the effects that were obtained by the repeated oral and the single i.a. injection of compounds **4** and **5** was performed.

The compounds analysed in the study are the leaders of a series of new synthesized molecules previously tested against articular pain; after a single *per os* administration, **4** and **5** were the most effective [13]. The compounds consisted of a NSAID fragment, sulindac (for **4**), and diclofenac (for **5**) (acting as COX inhibitors), linked to a sulphonamide-based CAI moiety. 

These have been demonstrated to be effective inhibitors of the CA isoforms I, II, IX, and XII and ineffective for the isoform IV [13]. 

Since the introduction of CAIs in the clinical use in the 40′, they still are the first choice for the treatment of oedema [16], altitude sickness, glaucoma [17], and epilepsy [18]. New advances are represented by the use of CAIs for the treatment of metabolic dysfunctions, including obesity [19], as well as the validation of the hCAs IX/XII as therapeutic targets for the treatment of solid hypoxic tumors [20]. Recently, proof-of-concept studies demonstrated that the targeting of hCAs might be useful in the management of neuropathic pain [21], cerebral ischemia [22], and inflammation diseases, such as rheumatoid arthritis [11,12]. CFA, once injected into the tibiotarsal joint, causes an inflammatory arthritis resembling to rheumatoid arthritis [23]. Inflammation that is induced by CFA begins on the third day after injection and it holds steady up to day 14, increasing pain sensitivity [24]. Repeated treatments with **4** and **5** were active in relieving pain behaviour, both 7 and 14 days after treatment, with no evidences of tolerance development that usually represents a common adverse effect in the management of chronic persistent pain [25,26]. In particular, compound **4** showed a higher efficacy with respect to the molecule **5**, which could be due to both the NSAID and the CAI fraction. Histologically, CFA injection led to an intense inflammatory state and swelling severely altering the joint morphology [27]. Chronic NSAID-CAI hybrid molecules did not enhance the morphological recovery modifying and reducing the articular inflammation state as well as the joint diameter. 

Furthermore, we also investigated the potential efficacy of a single i.a. injection of **4** and **5** in the same animal model of arthritis. Patients and their physicians often prefer the intra-articular therapy with lubricants or viscosifiers as hyaluronic acid (HA) [28], since the common treatment for arthritis requires long-term management with drugs that are usually poorly tolerated [29,30,31]. During the course of the pathology, the synovial fluid undergoes degradation that is similar to other tissues of the joint, which manifests as a decrease in the amount and the average molecular weight of HA [32], which is correlated with joint pain and functional impairment [33]. Injection of HA into the joint acts to restore intra-articular lubrication, consequently improving joint biomechanics. However, restoring viscosupplementation and lubrification is not enough to counteract the progression of arthritis. Several studies highlighted the correlation between pain and the dysregulation of pH in the synovium, or other tissues, of patients affected by arthritis [34,35,36]. These evidence strongly supported the idea of an intra-articular treatment with our NSAIDs-CAIs hybrid molecules in order to restore the normal pH condition of the i.a. joint space, in addition to the anti-inflammatory properties that were carried out by COX inhibitors. To perform the i.a. treatment, **4** and **5** compounds were reformulated in liposomes to prepare a solution in view of the intra-articular administration. The use of drug delivery systems may help to overcome both chemical and biopharmaceutical issues, and in particular liposomes have been extensively studied for such purposes. Liposomes are very versatile carriers that are able to formulate both hydrophilic and hydrophobic molecules, increasing the drug stability and bioavailability, being widely used for parental administration [37,38,39]. A single i.a. injection with **4** and **5** was performed on day 7, when the disease was already full-blown. Compound **5**, CAI linked with diclofenac, reduced the pain towards noxious and non-noxious mechanical stimuli, as well as improved the motor coordination. On the other hand, hybrid **4**, which was linked with sulindac, was ineffective in modulating pain sensitivity. The explanation for the different efficacy highlighted is not clear, which will surely require further investigations. 

As previously mentioned for the *per os* treatment, this protocol also did not improve the damage induced by CFA-injection at the articular level, as reported by the histological analysis. 

In conclusion, we deeply investigated new low molecular weight NSAID-CAI hybrid molecules as potential drugs for the treatment of ache-related symptoms that are associated with inflammatory diseases, such as the RA. In particular, we demonstrated the efficacy of a chronic repeated *per os* treatment as well as a single intra-articular treatment in the rat model of CFA-induced RA. NSAID-CAI hybrid molecules are suggested as a new pharmacological approach to treat persistent articular pain.

## 4. Materials and Methods

### 4.1. General Procedure for Synthesis of Compounds 4 and 5

All anhydrous solvents and reagents that were used in this study were purchased from Alfa Aesar, TCI, and Sigma-Aldrich. The synthetic reactions involving air- or moisture-sensitive chemicals were carried out under a nitrogen atmosphere using dried glassware and syringes techniques in order to transfer the solutions. More details are provided in the Appendix A. 

### 4.2. Animals

Sprague Dawley rats (Envigo, Varese, Italy) weighing 220–250 g at the beginning of the experimental procedure were used. Animals were housed in the Centro Stabulazione Animali da Laboratorio (University of Florence) and used at least one week after their arrival. Four rats were housed per cage (size 26 cm × 41 cm); animals were fed a standard laboratory diet and tap water ad libitum and kept at 23 ± 1 °C with a 12 h light/dark cycle (light at 7 A.M.). All of the animal manipulations were carried out according to the Directive 2010/63/EU of the European parliament and of the European Union council (22 September 2010) on the protection of animals used for scientific purposes. The ethical policy of the University of Florence complies with the Guide for the Care and Use of Laboratory Animals of the US National Institutes of Health (NIH Publication No. 85-23, revised 1996; University of Florence assurance number: A5278-01). Formal approval to conduct the experiments described was obtained from the Italian Ministry of Health (No. 517/2017, 06/04/2017) and from the Animal Subjects Review Board of the University of Florence. Experiments involving animals have been reported according to the ARRIVE guidelines [40]. All efforts were made to minimize animal suffering and to reduce the number of animals used. 

### 4.3. Complete Freund’s Adjuvant-Induced Rheumatoid Arthritis

Articular damage was induced by the injection of complete Freund’s adjuvant (CFA; Sigma-Aldrich St Louis, MO, USA), containing 1 mg/mL of heat-killed and dried Mycobacterium tuberculosis in paraffin oil and mannide monooleate, into the tibiotarsal joint [41,42]. More details are provided in the Appendix A. 

### 4.4. Repeated per os Treatment with Compounds 4 and 5

The compounds were suspended in a 1% solution of carboxymethylcellulose sodium salt (CMC) and *per os* daily administered, both at the dose of 1 mg kg^−1^, starting from the day of CFA intra-articular (i.a.) injection. Behavioural measurements were conducted on days 7 and 14, 30 min and 24 h after the last treatments with 4 and 5.

### 4.5. Materials for Liposomes Preparations

All of the solvents used were of High Performance Liquid Chromatography (HPLC) grade; Acetone, EtOH, CH_3_CN and CH_2_Cl_2_ were purchased from Merck (Darmstadt, Germany). HCOOH (85%) was provided by Carlo Erba (Milan, Italy). Water was purified by a Milli-Q plus system from Millipore (Milford, MA, USA). Natterman Phospholipids, GmbH kindly supplied egg phosphatidylcholine (Phospholipon® 90, P90) (Cologne, Germany). Cholesterol (CHOL) was purchased from Aldrich (Milan, Italy). 

### 4.6. HPLC PDA Analysis

The HPLC system consisted of a HP 1100 L instrument with a Diode Array Detector and was managed by a HP 9000 workstation (Agilent Techologies, Palo Alto, CA, USA). More details are provided in the Appendix A. 

### 4.7. Preparation of Liposomes 

Empty and loaded liposomes were prepared according to the film hydration method [37,38]. More details are provided in the Appendix A. 

### 4.8. Characterization of Liposomes

All of the developed nanocarriers were characterized in term of size, polydispersity, and ζ-potential by photon correlation spectroscopy. For these measurements, each liposomal dispersion was diluted 200-folds with deionized water. More details are provided in the Appendix A.

### 4.9. Intra-Articular Treatment with Lipo **4** and Lipo **5**

Fifty µL of each liposomial formulation were intra-articular (i.a.) injected seven days after CFA. The behavioural measurements were conducted on days 7 and 14 after lipo **4** and lipo **5** administration. 

### 4.10. Paw-Pressure Test

An analgesimeter determined the nociceptive threshold of rats (Ugo Basile, Varese, Italy), according to the method that was described by Leighton et al. [43]. 

### 4.11. Von Frey Test

The animals were placed in 20 × 20 cm plexiglass boxes equipped with a metallic meshy floor, 20 cm above the bench. The animals were allowed to habituate themselves to their environment for 15 min before the test [24,44]. More details are provided in the Appendix A. 

### 4.12. Beam Balance Test

The beam balance test was performed to assess the ability of the animal to remain upright and to walk on an elevated and relatively narrow beam. More details are provided in the Appendix A. 

### 4.13. Histological Evaluation

Animals were sacrificed by cervical dislocation. The legs were cut under the knee, flayed, and fixed in 4% formaldehyde in PBS for 48 h at room temperature. Subsequently, samples were decalcified by treatment with 0.76 M sodium formiate, 1.6 M formic acid solution in H2O for four weeks with a change of solution every seven days. At the end of the decalcification, these samples were dehydrated in alcohol and then embedded in paraffin. Sections (6 µm thick) of the tibiotarsal joint were haematoxylin and eosin stained and qualitatively analyzed by two independent observers in a blind fashion. Several morphological parameters (inflammatory infiltrate, fibrosis, bone, and cartilage erosion) were observed and quantified by a specific score (0: absent; 1: light; 2: moderate; 3: severe) according to Snekhalatha, et al. [45]. 

### 4.14. Statistical Analysis

The behavioural measurements were performed on eight rats for each treatment carried out in two different experimental sets. Results were expressed as mean (S.E.M.) with one-way analysis of variance. A Bonferroni’s significant difference procedure was used for a post hoc comparison. *p*-values <0.05 or <0.01 were considered to be significant. The data were analysed using the Origin 9 software (OriginLab, Northampton, MA, USA). Two sections for each animal were analysed for histological qualitative evaluations.

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
