# Peer review of "Pain Relieving Effect of-NSAIDs-CAIs Hybrid Molecules: Systemic and Intra-Articular Treatments against Rheumatoid Arthritis"

_ijms, 2019, doi:10.3390/ijms20081923_

Round 1

Reviewer 1 Report

Very interesting work! I suggest to publish it almost as it is. Even though there is given reference to previous work (ref. 13) where compounds 4 and 5 suppose to be described, depiction of exact  structure of these compounds is difficult. Therefore, structures of both compounds 4 and 5 studied in the work has to be added. 

Author Response

Response to Reviewer 1 Comments

Point 1. Very interesting work! I suggest to publish it almost as it is. Even though there is given reference to previous work (ref. 13) where compounds 4 and 5 suppose to be described, depiction of exact  structure of these compounds is difficult. Therefore, structures of both compounds 4 and 5 studied in the work has to be added.

Response. We thanks the referee for the revision and for appreciate our work. We agree with the reviewer and we have added in red the exact structure of compounds 4 and 5 in the main test.

Reviewer 2 Report

In this manuscript the authors reported an in vivo analysis of NSAIDs-CAIs hybrid molecules previously reported by the same research group.

In my opinion, it is an interesting work with relevant results. Usually, when a manuscript represents a continuation of another paper, the second paper is somewhat less important because it lacks of novelty. However, this is not the case because this paper add important new in vivo results that highlight a real possible use of NSAIDs-CAIs hybrid molecules.

There is only some minor comments:

I suggest to remove the word “new” in the title.

A picture representing compounds 4 and 5 (including their activity on the different CAs) would be very useful for the readers.

As the compounds have been efficiently tested in vivo I suggest to introduce a short paragraph reporting on in silico ADME (a logP is enough, in order to better classify these  compounds).

Author Response

Response to Reviewer 2 Comments

In this manuscript the authors reported an in vivo analysis of NSAIDs-CAIs hybrid molecules previously reported by the same research group.

In my opinion, it is an interesting work with relevant results. Usually, when a manuscript represents a continuation of another paper, the second paper is somewhat less important because it lacks of novelty. However, this is not the case because this paper add important new in vivo results that highlight a real possible use of NSAIDs-CAIs hybrid molecules.

There is only some minor comments:

Point 1. I suggest to remove the word “new” in the title.

Response. The word “new” has been removed from the title

Point 2. A picture representing compounds 4 and 5 (including their activity on the different CAs) would be very useful for the readers.

Response. A picture representing the molecular structure of compounds 4 and 5 has been included in the results section of the main test. Moreover we provided a table regarding the activity of these compounds on the different CAs. The changes made are highlighted in red.

Point 3. As the compounds have been efficiently tested in vivo I suggest to introduce a short paragraph reporting on in silico ADME (a logP is enough, in order to better classify these  compounds.

Response. As suggested by the reviewer, we added the logP values of both compounds in the results section. The modifications made are highlighted in red.